

**The relative impact of cloud condensation nuclei and ice nucleating**
**particle concentrations on phase-partitioning in Arctic Mixed-Phase**
**Stratocumulus Clouds**
Amy Solomon[1,2], Gijs de Boer[1,2], Jessie M. Creamean[1,2,a], Allison McComiskey[2], Matthew
D. Shupe[1,2], Maximilian Maahn[1,2], and Christopher Cox[1,2]
(1) Cooperative Institute for Research in Environmental Sciences, University of Colorado
Boulder, Colorado, USA.
(2) Earth System Research Laboratory, National Oceanic and Atmospheric Administration,
Boulder, Colorado, USA.
[a]Now at: Department of Atmospheric Sciences, Colorado State University, Fort Collins,
Colorado, USA.
Corresponding author: Amy Solomon, NOAA/ESRL, PSD3, 325 Broadway, Boulder,
Colorado 80305-3337, USA. (amy.solomon@noaa.gov)

**Abstract**
This study investigates the interactions between cloud dynamics and aerosols in idealized
large-eddy simulations of an Arctic mixed-phase stratocumulus cloud observed at Oliktok
Point, Alaska in April 2015. This case was chosen because it allows the cloud to form in
response to radiative cooling starting from a cloud-free state, rather than requiring the cloud
ice and liquid to adjust to an initial cloudy state. Sensitivity studies are used to identify whether
there are buffering feedbacks that limit the impact of aerosol perturbations. The results of this
study indicate that perturbations in ice nucleating particles (INPs) dominate over cloud
condensation nuclei (CCN) perturbations, i.e., an equivalent fractional decrease in CCN and
INPs results in an increase in the cloud-top longwave cooling rate, even though the droplet
effective radius increases and the cloud emissivity decreases. The dominant effect of ice in the
simulated mixed-phase cloud is a thinning rather than a glaciation, causing the mixed-phase
clouds to radiate as a grey body and the radiative properties of the cloud to be more sensitive
to aerosol perturbations. It is demonstrated that allowing prognostic CCN and INP causes a
layering of the aerosols, with increased concentrations of CCN above cloud top and increased
concentrations of INP at the base of the cloud-driven mixed-layer. This layering contributes to
the maintenance of the cloud liquid, which drives the dynamics of the cloud system.



## 1 Introduction

Arctic mixed-phase stratocumulus clouds (AMPS) play a unique role in climate by producing a net warming at the Earth's surface over the annual cycle. This warming is due to the limited amount of incoming solar radiation at high latitudes, causing downward longwave radiative effects to dominate surface cloud forcing in the Arctic. AMPS are characterized by a liquid cloud layer with ice crystals that precipitate from cloud base even at temperatures well below freezing (Hobbs and Rangno, 1998; Intrieri et al., 2002; McFarquhar et al., 2007). The magnitude of the cloud-forced surface warming is primarily a function of the liquid water content of the AMPS and the properties of the cloud droplets (Curry and Ebert 1992; Curry et al. 1993; Zhang et al. 1996) However, different from warm clouds, the magnitude and properties of cloud liquid in mixed-phase clouds are closely connected to the formation of cloud ice, which limits the availability of water vapor for droplet formation and growth (the Wegener-Bergeron-Findeisen (WBF) mechanism, Wegener, 1911; Bergeron, 1935; Findeisen, 1938) and acts as a sink for water vapor through the growth and sedimentation of frozen precipitation.

Cloud ice in AMPS must form through heterogeneous nucleation, since temperatures are too warm for homogenous ice nucleation (approximately > 36 ˚C). Heterogeneous ice nucleation can occur by a number of modes: either in the presence of super-cooled droplets, when an aerosol comes into contact with a droplet (contact freezing), is immersed in a droplet followed by freezing (immersion freezing), or in the absence of droplets through vapor deposition on aerosol (deposition freezing) or liquid forming on aerosol (condensation freezing) (Pruppacher and Klett, 1997). The efficiency of any of these modes in a given environmental state is a function of aerosol properties, which determine whether an aerosol can serve as an ice nucleating particle (INP), cloud condensation nucleus (CCN), or both. Based on measurements from in situ instrumentation and the reduced concentration of ice crystals relative to liquid droplets (Murray et al., 2012), it is believed that only a small fraction of aerosols can serve as INPs. For example, in the Indirect and Semi- Direct Aerosol Campaign (ISDAC) (McFarquhar et al., 2011) that took place off the coast of Utqiaġvik, (Barrow) Alaska in the spring of 2008, the number of INPs available were observed to be four orders of magnitude smaller than the



number of aerosols serving as CCN. However, it is important to note that only a few INPs are
needed to glaciate a cloud (see discussion in DeMott et al. 2010).
The impact of CCN variability on warm cloud structure has been the subject of numerous
papers. For a fixed liquid water path (LWP), an increase in CCN in warm clouds will increase
the number of droplets and reduce the droplet size. This causes an increase in cloud albedo (the
Twomey effect; Twomey, 1977) and potentially suppresses precipitation (the Albrecht effect;
Albrecht 1989). Suppressing precipitation can increase cloud thickness/coverage and cloud-
driven turbulence, which increases the entrainment of dry air at cloud top, thereby thinning the
cloud (Pincus and Baker, 1994; Stevens et al., 1998). This thinning of the cloud as a response
to an increase in aerosols is an example of a buffering feedback that limits the impact of a CCN
perturbation on the cloud structure (Stevens and Feingold, 2009). For LWPs greater than
approximately 50 gm$^{-2}$, AMPS emit as blackbodies (Shupe and Intrieri, 2004) and an increase
in CCN has minimal impact on longwave emissivity (Morrison et al., 2008). However, smaller
droplets can also reduce the ice water path (IWP) through a reduction in collision-coalescence
and riming of snow by droplets (Morrison et al., 2008), as well as, make ice nucleation less
efficient (Lance et al. 2011). These are only a few examples of buffering feedbacks that exist
in mixed-phase clouds.
An increase in INPs in AMPS is known to produce a "glaciation effect", i.e., a rapid depletion
of cloud liquid, in part due to the WBF mechanism and the acceleration of frozen precipitation
(Murray et al., 2012). However, the efficacy of this effect is dependent upon the chemical
composition of the INPs. In cases where INPs are transported over long distances and coated
in sulfate or organic materials, an increased concentration of INPs may actually be linked to a
"deactivation effect". This is because coated particles generally freeze at colder temperatures
(Cziczo et al., 2009; Sullivan et al., 2010; Girard and Sokhandan, 2014), changing the number,
size and fall speeds of nucleated ice crystals and increasing cloud lifetime. The deactivation
effect can cause a significant increase in surface warming, since a decrease in droplet size in
optically thin clouds for constant LWP can produce a significant increase in cloud longwave
emissivity (Garrett and Zhao, 2006).



Given the uncertainty in measurements of INPs (DeMott, 2015; Garimella et al., 2017), it has
been challenging to study the manifestation of aerosol-cloud interactions in mixed-phase cloud
conditions. This study investigates these interactions and their dependence on aerosol
partitioning in idealized large-eddy simulations (LES) of an AMPS observed at Oliktok Point,
Alaska 5-17Z 17 April 2015. In order to isolate the impact of these interactions on longwave
cloud forcing, shortwave radiation is neglected in the simulations. The microphysics used in
Solomon et al. (2015) have been modified to include prognostic CCN in addition to prognostic
INPs. This allows for a more realistic representation of aerosols advected over the Oliktok
Point site. In order to identify aerosol indirect effects in these simulations, aerosol chemistry
is specified and, due to the in-cloud temperatures for this case, all cloud ice forms through
immersion freezing. The focus on immersion freezing is supported by studies demonstrating
that liquid droplets typically form prior to ice formation in mixed-phase cloud environments
(e.g. de Boer et al., 2011; Westbrook and Illingworth, 2012).
This study is focused on the research questions:
1) What is the relative impact of CCN versus INP perturbations on the phase partitioning

16       between cloud liquid and cloud ice in AMPS? Specifically, what is the impact on cloud

17       dynamics?

2) Are there buffering feedbacks in AMPS that limit the impact of CCN/INP

19       perturbations?

**2  Case Description**
Simulations are set up to recreate conditions observed by the US Department of Energy (DOE)
Atmospheric Radiation Measurement (ARM) program third mobile facility (AMF-3) at
Oliktok Point, Alaska on and around 17 April 2015 (Figure 1). During this time period, Oliktok
Point was situated in a relatively quiet regime synoptically, with surface high pressure to the
northwest over the Chukchi Sea, and a weak area of low pressure over central Alaska. This
resulted in steadily increasing surface air pressure from around 1008 hPa at 00Z on 16 April
to around 1021 hPa at 00Z on 18 April and relatively light (2-5 m s$^{-1}$) near-surface winds ($U_{10m}$)
from the east-northeast over this entire two-day window (Figures 1f,g). Near-surface air
temperatures ($T_{2m}$) varied dramatically (Figure 1e), depending on time of day and cloud cover,



with the coldest temperatures around 250 K during late evening and early morning clear
periods on 16 April, and the warmest temperatures being around 258 K during cloudy periods
during local night time on 17 April. Near-surface relative humidity ($RH_{2m}$) showed a weak
diurnal signature, ranging from around 80% during local daytime hours and 86-88% during
nighttime hours.
Radiosonde launches conducted at approximately 2330 UTC on 15 April, 1730 UTC and 2330
UTC on 16 April, and 1830 UTC and 2330 UTC on 17 April reveal the evolution of the lower
troposphere at Oliktok Point (Figure 1h). The 2330 UTC sounding from 15 April reveals a
well-mixed surface layer extending up to around 200 m, a stable layer from 200-700 m and
then an elevated layer that is generally well mixed from 700-2000 m. By 1730 UTC a strong
inversion formed from 30-300 m above the surface where there is a residual mixed layer from
the previous night's cloud cover between 300-600 m. Above this layer, there is a weakly stable
layer from 600 m to around 1000 m. A similar vertical structure persists through the rest of
the sampling period, with the strength of the near-surface inversion and depth of the overlying
mixed layer evolving with time, and general warm-air advection occurring above 1000 m.
With the exception of a very dry layer between 200-350 m in the 2330 UTC sounding on 17
April, and a saturated layer around 200 m in the 1730 UTC sounding on 16 April, the lower
atmosphere (0-700 m) features relative humidity between 70-90%. Above this, RH values
feature variability between 30-60%, with the height at which this drop off occurs increasing
with time. The last two soundings are substantially drier between 1200-2000 m, with RH
values around 20%.
On both 16 and 17 April, low-level mixed-phase stratus cloud layers were observed to develop
during local night time (~0600-1800 UTC). Surface radiation measurements (Figures 1c,d)
clearly demonstrate the presence of these clouds, with the net longwave radiation ($LW_{NET}$)
increasing to around 0 W m$^{-2}$ as a result of increased downwelling longwave ($LW_{DOWN}$) during
cloudy periods. An increase in downwelling shortwave radiation may be playing a role in the
dissipation of the cloud layer. The presence of cloud is also detected by active remote sensors
(Figure 1a), with ceilometer cloud base (black dots) and Ka-band cloud radar reflectivity
(colored shading) measurements clearly showing the nighttime appearance of liquid-
containing cloud layers between 450 and 700 m (on 16 April) and 800 and 1000 m (on 17



April) above the surface, with the cloud on 17 April starting and ending with lower cloud bases
and tops.  Both radar and surface precipitation gauges indicated weak snowfall associated with
these clouds and shortwave irradiance measurements reveal that the surface was snow covered
during this time, with surface albedo around 80%.

## 3    Model Description

Simulations are completed using the large eddy simulation mode of the Advanced Research
WRF model (WRFLES) version 3.3.1 (Yamaguchi and Feingold, 2012) with the RRTMG
longwave radiation parameterization (Mlawer et al., 1997) and the Morrison two-moment
microphysical scheme (Morrison et al., 2009). Collision-coalescence was found to be
important in CCN perturbation studies during the Fall 2004 Mixed-Phase Arctic Cloud
Experiment (M-PACE; Morrison et al, 2008), however tests with and without riming and
collision-coalescence indicated that these processes are not significant for this case and have
been neglected in the simulations. Shortwave radiation is neglected given the nighttime
occurrence of these clouds and to be able to focus on longwave indirect aerosol effects. Surface
fluxes are calculated using the modified MM5 similarity scheme, which calculates surface
exchange coefficients for heat, moisture, and momentum following Webb (1970) and uses
Monin–Obukhov with Carlson–Boland viscous sublayer and standard similarity functions
following Paulson (1970) and Dyer and Hicks (1970). The land surface is simulated with the
unified Noah land-surface model (Tewari et al., 2004). Initial surface pressure is 1020 hPa.
The initial surface temperature is 255 K.
All simulations are run on a domain of $3.6 \times 3.6 \times 1.4$ km with a horizontal grid spacing of 50
m and vertical spacing of 10 m. The domain has $72(x) \times 72(y) \times 140(z)$ grid points and is
periodic in both the x and y directions. The top of the domain is at 1.4 km. The model time
step is 0.5 seconds.

## a.  Initial atmospheric profiles

Initial profiles of temperature, moisture and horizontal wind components are based on
radiosonde measurements taken at Oliktok Point at 23:30Z 16 April 2015 (Figure 1h).
Soundings are not available during the cloudy period (5-18Z), seen in the KAZR reflectivity



(Figure 1a). Therefore, the model is initialized with a cloud free sounding and the cloud forms
in response to the radiative cooling, rather than starting the stimulation with a "cloudy" profile
and requiring the cloud ice and liquid to adjust to the initial state. The potential temperature
and water vapor mixing ratio from the radiosonde and the initial profiles used in the simulations
are shown in Figure 2. Initial water vapor is increased in the region where cloud liquid water
was observed after 5Z in order to produce cloud liquid water at the start of the integration.
Initial temperature and subgrid TKE are perturbed below the top of the mixed layer with
pseudo-random fluctuations with amplitude of 0.1 K and 0.1 $m^2$ $s^{-2}$, respectively.
## *b. Large-scale forcing*
Large-scale subsidence is specified by integrating the prescribed horizontal wind divergence
from the surface upward. Divergence is assumed to be equal to $1.8 \times 10^{-6}$ $s^{-1}$ below the
inversion and zero above, following the Solomon et al. (2015) study. This gives a linear
increase in large-scale subsidence from zero at the surface to 1.5 mm $s^{-1}$ at the base of the
initial inversion ($z$=805 m), above which the large-scale vertical wind is constant. Large-scale
subsidence is accounted for via a source term for any prognostic variable other than wind
components.
Temperature and moisture profiles are nudged to the initial profiles in the top 100 meters of
the domain with a time scale of 1 hour. Horizontal winds are nudged to the initial profiles at
and below the initial inversion base with a timescale of 2 hours. Nudging of the horizontal
wind components, temperature and moisture profiles is performed by adding a source term to
the prognostic equations for potential temperature, water vapor, and horizontal wind
components.
## *c. Droplet number concentration and CCN properties*
Because CCN measurements were not available from Oliktok Point during this time, initial
CCN size distributions at every gridpoint are based on springtime measurements taken during
the ISDAC campaign (Earle et al. 2011). The accumulation mode observed during ISDAC had
a concentration of less than 200 $cm^{-3}$ (165 $cm^{-3}$ used in this study), a modal diameter of 0.2
microns, and a geometric standard deviation of 1.4. Sensitivity studies vary initial CCN



concentration with an arbitrary multiplication factor *C* (referred to as the CCN factor). CCN
mean concentration is then treated as a prognostic variable. A prognostic equation for CCN
number concentration has been added to WRFLES,

$$\frac{\partial CCN}{\partial t} + ADV + DIFF = \frac{\delta CCN}{\delta t}\bigg|_{condensation} + \frac{\delta CCN}{\delta t}\bigg|_{evaporation} \qquad (1)$$

where ADV represents advection and DIFF represents turbulent diffusion. Condensation is a
sink of CCN and evaporation is a source of CCN. Evaporation (condensation) of one droplet
produces (removes) one CCN.
Cloud droplets are activated using resolved and subgrid vertical motion (Morrison and Pinto,
2005) and a log-normal aerosol size distribution (assumed to be ammonium bisulfate and 30%
insoluble by volume) to derive cloud condensation nuclei spectra following Abdul-Razzak and
Ghan (2000). As noted in Solomon et al. (2015), because the aerosol number size distribution
peaks at a relatively large diameter of 0.2 microns, the majority of CCN activate into droplets
at low supersaturation (at or below $SS_w = 0.15\%$) for a reasonable range of aerosol composition
assumptions. Since such supersaturations can be generated even by slow updrafts, the
sensitivity of droplet number concentration to aerosol composition is expected to be weak. We
therefore only include simulations that test the sensitivity to mean CCN concentrations.
### *d. Ice nucleation*
Following Solomon et al. (2015), a prognostic equation for INP number concentration ($N_{INP}$)
has been added to WRFLES,

$$\frac{\partial N_{INP}}{\partial t} + ADV + DIFF = \frac{\delta N_{INP}}{\delta t}\bigg|_{activation} + \frac{\delta N_{INP}}{\delta t}\bigg|_{sublimation} \qquad (2)$$

where ADV represents advection and DIFF represents turbulent diffusion. Activation is also
referred to as ice nucleation and sublimation represents a source of INP, supporting the
recycling of these particles.



Eight prognostic equations are integrated for $N_{INP}$ in equally spaced temperature intervals with
nucleation thresholds spanning the range of the in-cloud temperatures (-20.15°C to -14.55°C).
Therefore, additional INP become available for activation with decreasing temperature and as
the cloud layer cools. INP number concentrations are initially specified at 1.3 L$^{-1}$ in each bin
to produce IWP close to observations for this case. Sensitivity studies vary initial INP
concentration with an arbitrary multiplication factor $F$ (referred to as the INP factor).
It is assumed that 50% of the INP available in a bin nucleate if the in-situ temperature is below
the threshold temperature and the local conditions exceed water saturation. Therefore, initial
$N_{INP}$ are a function of the nucleation threshold temperatures and are independent of the in-situ
temperature. The in-situ temperature in regions of water saturation determines how many INP
are activated. Due to the pristine dendritic nature of the observed crystals and the limited
number of INP, ice shattering and aggregation are neglected in the simulations and sublimation
returns one INP per sublimated crystal.
$N_{INP}$ (in units of L$^{-1}$) integrated over the domain in each temperature bin $k$ at time $t$ is equal
to

$$\overline{N}_{INP}(k,t) = \iiint N_{INP}(x,y,z,k,t)\ dx\, dy\, dz. \tag{3}$$

Upon sublimation, the modification of activation thresholds that can occur for previously
nucleated INP, i.e. preactivation (Roberts and Hallett, 1967), is not considered and $N_{INP}$ are
returned to each bin $k$ with weighting

$$W_k = [\,\overline{N}_{INP}(k,0) - \overline{N}_{INP}(k,t)\,]\,/\,\overline{N}_{INP}(k,0) \tag{4}$$

where $W_k$ is normalized such that $\sum W_k = 1$. The $W_k$ are recalculated each time step. In this
way, INP are recycled preferentially to each of the eight temperature bins from which they
originated (Feingold et al., 1996).
**4   Simulations Completed**



Simulations completed for this study are listed in Table 1. A simulation with INP and CCN
factors equal to 1.0 is referred to as the Control. To isolate the impact of CCN perturbations
on mixed-phase clouds without a change in ice formation, three simulations were run with
CCN factors 1/2, 1, 2 and INP factors equal to 1.0 (runs ConIce0.5, Control, ConIce2.0). To
identify the impact of CCN perturbations on mixed-phase clouds when ice formation is a linear
function of the CCN number concentration, two runs were done with INP and CCN factors
equal to 2, one with fixed CCN (FixedCCN2.0) and one with prognostic CCN (LinIce2.0), and
one run with INP and CCN factors equal to 1/2 (LinIce0.5).
## 5  Model Results
### *a. Control simulation*
The control simulation (CCN and INP factors equal to one) has IWP and LWP consistent with
observations on 17 April 2015 (Figures 1b and 3a), and a cloud system that reaches a steady
state after 5 hours with liquid water fractions close to 0.5. The cloud-driven mixed-layer depth
slowly increases over the 16-hour integration, with both cloud top rising and cloud base
lowering at a rate of ~5 m/hour. However, the cloud system remains decoupled from the
surface layer and a surface inversion of ~180 meters in depth is maintained throughout the
integration.
Rain forms in the liquid layer (with concentrations less than 0.02 cm$^{-3}$) and evaporates within
200 meters of cloud base. Therefore, the production and impact of rain on this simulation can
be neglected. This is true for the sensitivity studies discussed below as well but would not be
the case for runs with more limited CCN, which would produce significant precipitation.
Snow does reach the surface during the steady state with a relatively constant flux of $6 \times 10^{-4}$ g
m$^{-2}$ s$^{-1}$. Sublimation at the base of the cloud-driven mixed-layer reduces the snow water content
by $6 \times 10^{-6}$ g m$^{-3}$ s$^{-1}$. In terms of number concentration, this causes the recycling of
approximately 1 L$^{-1}$ hour$^{-1}$ of INP back into the cloud-driven mixed-layer. Since total ice
crystal number concentrations are 1-2 L$^{-1}$ in the mixed-layer over the integration, this indicates
that recycling of INP is playing a significant role in the maintenance of cloud ice in this cloud
system (e.g., Solomon et al. 2015).



While the cloud system is maintaining a steady state in cloud ice and liquid, longwave cooling
continually cools the cloud-driven mixed-layer, contributing to the maintenance of the phase-
partitioning by increased activation of INP within the liquid cloud layer and depletion of water
vapor within the mixed-layer. This cooling is required to maintain the cloud liquid because of
the continuous depletion of water vapor, and to maintain the cloud ice, since ~2 L$^{-1}$ of INP are
lost to the surface through precipitation each hour.
### b. Impact of CCN perturbations with constant ice formation
The first set of simulations completed for this study tests the sensitivity of the cloud to
perturbations in CCN concentrations, while keeping INP concentrations fixed (ConIce
simulations). Simulations involved increasing and decreasing the initial CCN concentrations
by a factor of two. These runs provide insight into the impact of CCN perturbations in an Arctic
environment with stable stratification near the surface and a weak inversion at cloud top with
relatively moist air. Figure 3a shows that increasing the CCN concentrations has the expected
effect at the beginning of the integrations when the cloud layer is optically thin-- smaller
droplets increase the cloud emissivity and thereby longwave cooling at cloud top, increasing
turbulence, supersaturation, and droplet formation. The opposite result is found with
decreasing CCN concentrations. For these optically thin cases, radiative cooling is a
contribution from the full physical depth of the cloud and the increased cooling rate is therefore
an expression of increased total emission of the cloud. Figure 3b shows that this also results in
faster vapor deposition rates, resulting in increased ice mass given an increase in CCN.
However, this increase in IWP plateaus and slowly decreases after hours 5-6 as ice crystals fall
out of the cloud layer and are lost from the system. Decreasing CCN results in slower
deposition rates and reduced IWP.
What is unexpected is that the minimum longwave heating rate (or maximum cooling rate) is
not exclusively a function of LWP, even after the clouds become optically thick (Figure 3c).
The larger longwave cooling rate associated with a cloud forming in conjunction with elevated
CCN concentrations causes increased total water in the cloud-driven mixed-layer (indicating
that liquid water increases faster than ice mass), which increases the water vapor mixing ratio
below the liquid layer, increasing relative humidity in the mixed-layer below the liquid layer



(see Figure 4a). The IWP shows larger temporal variability than the LWP, potentially due to
larger sensitivity to small perturbations in relative humidity.
Increasing CCN causes the cloud top to rise and the cloud base to lower faster while keeping
the maximum buoyancy relatively unchanged, the maximum buoyancy as a function of LWP
is similar for ConIce2.0 and Control, with Control approximately $1 \times 10^{-4}$ less than ConIce2.0
for LWP less than 17 $gm^{-2}$  (Figure 3d). The deepening of the cloud layer for an increase in
CCN causes the impact of CCN on maximum droplet size to persist throughout the integration
(Figure 5a), even though LWP is increasing more rapidly (Figure 3a).
The ConIce set of simulations was designed to identify the impact of CCN variability on phase-
partitioning and cloud dynamics for a relatively constant magnitude of cloud ice number
concentration. This is seen to be the case in Figure 5c, however, interestingly, processes such
as sublimation cause ConIce2.0 to have less total cloud ice number concentration than the other
two runs (see Figure 4a-d), since a reduction in sublimation causes more INP to fall out of the
mixed-layer and less recycling of INP in the cloud layer. However, the differences in relative
humidity also play a role here and therefore the sublimation rate is not just a function of the
ice present. Increasing CCN causes more rapid mixed-layer cooling (Figure 4b and 5b) and
deeper mixed-layer depths, and therefore larger net deposition rates (Figure 5d) and larger
IWPs (Figure 3b).
*c. Impact of CCN perturbations with linear ice formation*
A second set of simulations was completed adding an additional degree of realism to the
simulations by scaling INP concentrations with the CCN concentration, i.e., equal INP and
CCN factors (LinIce simulations). This was done in order to represent the case of a polluted
airmass with equal relative increases of CCN and INP.  The fraction of INP to CCN evolves
in time as cloud ice forms and precipitates, sublimates below the cloud base, and advects back
into the cloud layer with the cloud-driven vertical motions.
Figure 6 shows the sensitivity of the LWP and IWP to an increase and decrease in CCN and
INP by a factor of 2. Similar to the ConIce runs, the increase in LWP begins to slow when IWP
exceeds 20 g $m^{-2}$. Before hour 5 the ConIce and LinIce runs have similar trajectories. After



hour 5 these runs diverge; decreasing CCN and INP factors in LinIce results in larger LWP because of the dominating effect of INP variability on water vapor availability. Increasing CCN and INP factors from 0.5 to 1.0 results in a 63% decrease in LWP at hour 15.

The LinIce runs highlight the extreme sensitivity of phase-partitioning between cloud liquid and ice to very small changes in INP concentrations. For these simulations, there are 5 orders of magnitude difference between CCN and INP concentrations, with the scaling done on a percentage basis relative to the absolute amount. In other words, a change in INP from 1 to 2 $L^{-1}$ has a substantially larger effect on cloud phase than a change in CCN from 10,000 to 20,000 $L^{-1}$, despite the latter change obviously being much more extreme from the perspective of aerosol number concentration. The CCN variability in the LinIce runs cause a 50% increase in droplet effective radius for a decrease in CCN by a factor of two (Figure 7a). These changes are of the same sign as the ConIce runs but are larger due to the additional impact of INP variability on water vapor available for condensation. Similarly, the reduction of cloud droplet size related to the presence of elevated aerosol concentrations is exacerbated when both INP and CCN concentrations increase due to the combination of more nucleated droplets (on CCN) and increased water vapor deposition (on ice crystals generated by the elevated INP concentrations). However, the impact of the changes in droplet effective radius are small relative to the impact of INP variability on the dynamics of the cloud-driven mixed layer after approximately hour 4. For example, after hour 7 the LinIce0.5 run has a colder cloud-driven mixed-layer than the Control and LinIce2.0 runs (Figure 7b), whereas ConIce2.0 had a colder cloud-driven mixed-layer than ConIce0.5 (Figure 5b).

The ratio of INP among the LinIce runs stays relatively constant for the 16-hour integrations (Figure 7c), indicating that differences in sublimation (INP recycling) and ice fall speeds do not produce appreciably different ice crystal number concentrations in the cloud-driven mixed-layer. However, larger deposition rates in the cloud-driven mixed-layer due to increased ice number concentrations (Figure 7d) result in larger ice water mixing ratios (a 31% increase between ConIce2.0 and LinIce2.0), even though mixed-layer temperatures are warmer for an increase in CCN and INP factors, indicating that NDEPS is dominant relative to ice number in controlling IWP.



Allowing both prognostic CCN and INP reveals interesting layering of aerosol distributions
that results from differential liquid and ice processes. This is seen in Figure 8, where the
turbulent advection of droplets into the cloud top inversion causes larger CCN concentrations
above the liquid cloud layer than below (Figure 8a). The opposite result is found for INP, where
gravitational settling of ice crystals and the subsequent sub-cloud sublimation produces
locally-elevated INP concentrations in the lowest 200 meters (Figure 8b). This layering of
aerosols causes larger entrainment of CCN at cloud top and larger entrainment of INP at mixed-
layer base as the mixed-layer deepens. As was demonstrated in Solomon et al (2015), the
increase in INP at the base of the mixed-layer contributed to the maintenance of the phase-
partitioning by making more INP available for activation as ice crystals as the mixed-layer
deepens.
This layering of aerosols, with increased CCN above the liquid layer and increased INP below
the cloud-driven mixed-layer, causes interesting differences between runs with and without
prognostic CCN (Figure 9). The increase in CCN above the liquid layer causes increased
entrainment of CCN at cloud top (Figure 9a), decreasing the droplet effective radius, increasing
the longwave cooling (Figure 9b), increasing the deposition rate and ice water mixing ratio
(Figure 9c).  Ultimately, these differences result in stronger buoyancy with the cloud layer
(Figure 9d), where buoyancy is increased by up to 10%.
**6  Summary and Discussion**
In this study we use idealized large-eddy simulations to quantify the relative impact of CCN
and INP perturbations on the phase-partitioning and dynamics of AMPS. The modeling
framework developed in Solomon et al. (2015) to study the recycling of INP has been extended
to include prognostic CCN. The first set of simulations were designed to investigate the impact
of relatively small perturbations in CCN compared to studies such as Morrison et al. (2008)
and Kravitz et al. (2014) in mixed-phase conditions with essentially constant INP on phase-
partitioning and cloud dynamics. It is found that increasing CCN by a factor of two increases
LWP by 60-100%, while decreases CCN by a factor of reduces LWP by less than 8%, i.e., the
impact is highly non-linear.  This change stems primarily from an increase in cloud longwave
emissivity and associated increase of cloud-top cooling rates, connected to a decrease in



droplet size.  This elevated cooling rate causes the cloud-driven mixed-layer to cool and deepen
more rapidly.  However, interestingly, this difference persists even when the cloud radiates as
a blackbody, highlighting the influence of early changes to the system.
The sensitivity to CCN in this study is significantly larger than that found in the M-PACE case
study by Morrison et al. (2008) and the ISDAC case study by Kravitz et al. (2014). The M-
PACE and ISDAC case studies are useful as examples of the extreme range of conditions under
which mixed-phase clouds exist. M-PACE took place in October over open water with large
fluxes of heat and momentum from the surface into the cloud layer, while ISDAC took place
during the spring when the Arctic Ocean was essentially ice-covered and the cloud layer was
decoupled from the surface layer.
In the M-PACE case study increasing CCN by a factor of 5-6 resulted in an increase in LWP
by 20% and a decrease in IWP by 60%. In the ISDAC case study, increasing CCN by a factor
of 4 resulted in similar LWP and IWP until hour 12 when the more pristine cloud collapsed. A
very interesting difference between this study and the two previous studies is that both the M-
PACE and ISDAC studies found IWP to decrease when CCN was increased, while this study
finds an increase in IWP due to increased longwave cooling and larger deposition rates. For
the ISDAC case study this may be due to the continuous decrease in droplet number
concentration, which would cause the cloud dynamics to spin down. This study also finds a
significantly larger sensitivity to INP than the M-PACE case study. The M-PACE case study
found that a 10X increase in INP resulted in a 120% increase in IWP and a 23% decrease in
LWP, while this study finds increasing INP by 2X results in a 60% increase in IWP and a 36%
decrease in LWP. These differences need to be investigated further to identify the relative
impact of different environmental conditions and model physics.
The differences between this study and the M-PACE and ISDAC studies are due in part to the
different environmental conditions but the different sensitivities to both CCN and INP are also
due to the different parameterizations used in these models. Both the M-PACE and ISDAC
studies essentially hold INP fixed, while this study does not constrain INP other than specifying
an initial value. This allows for the vertical redistribution of particles throughout the domain,




resulting in feedbacks between ice and droplet properties and cloud dynamics not included in
the two previous case studies and in many climate and weather-scale models.
Additionally, the inclusion of prognostic CCN and INP reveals a number of feedbacks that
buffer the cloud system from collapsing. The first buffering feedback is the recycling of INP,
which was the focus of the Solomon et al. (2015) study. The IWP, and therefore phase-
partitioning between cloud ice and liquid, cannot be maintained without this feedback.
However, the reservoir of INP below the mixed-layer would be depleted if cloud-driven mixed-
layer became coupled to the surface layer. The second interesting buffering feedback is the
layering of the aerosols, with increased concentrations of CCN above cloud top and increased
concentrations of INP at the base of the mixed-layer. As demonstrated in this paper, this
layering contributes to the maintenance of the cloud liquid, which ultimately drives the
dynamics of the cloud system. A third buffering feedback is the thinning of the liquid layer
when INP concentrations are increased. The occurrence of this thinning does not produce
complete glaciation, rather causes the mixed-phase clouds to radiate as grey bodies and the
radiative properties of the clouds to be more sensitive to aerosol perturbations.
Ultimately, enhanced observations of the vertical structure of cloud microphysics and aerosol
properties are required. Recent work by the DOE ARM program to operate unmanned aircraft
and tethered balloons has provided new perspectives on these quantities and the dynamic and
thermodynamic conditions supporting these cloud systems (de Boer et al. 2018). Such
observational efforts should continue to be pursued further to help constrain the sensitivities
demonstrated by numerical studies as presented here.
**7 Acknowledgments**
This research was supported by the US Department of Energy (DOE) Atmospheric Systems
Research (ASR) program (DE-SC0013306) and the National Oceanic and Atmospheric
Administration (NOAA) Physical Sciences Division (PSD). Additionally, measurements used
in this study were supported by the US Department of Energy Atmospheric Radiation
Measurement (ARM) program.





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

| Simulation Name | CCN Factor | INP Factor |
|---|---|---|
| --Constant Ice Runs-- | | |
| ConIce0.5 (red) | 1/2 | 1 |
| Control (black) | 1 | 1 |
| ConIce2.0 (blue) | 2 | 1 |
| --Linear Ice Runs-- | | |
| LinIce0.5 (red) | 1/2 | 1/2 |
| LinIce2.0 (blue) | 2 | 2 |
| FixCCN2.0 | 2 | 2 |

2 **Table 1:** Description of simulations completed and discussed in the paper.



## 10 Figure Captions

**Figure 1**: Measurements at Oliktok Point, Alaska 15-17 April 2015. (a) Ceilometer (black dots) and Ka-band cloud radar reflectivity (colored shading). B) LWP and IWP, in units of $gm^{-2}$. LW (c) and SW (d) surface radiation measurements, in units of $Wm^{-2}$. (e) Surface air temperatures. (f) Surface wind direction, in units of degrees. (g) Surface wind speed, in units of $ms^{-1}$. (h) Radiosondes at approximately 2330 UTC on 15 April, 1730 UTC and 2330 UTC on 16 April, and 1830 UTC and 2330 UTC on 17 April.

**Figure 2**: Sounding measured at 0 UTC 17 April 2015 at Oliktok Point, Alaska. Left: water vapor mixing ratio ($q_v$) and potential temperature (theta), in units of g $kg^{-1}$, and Kelvin respectively. Right: zonal wind (U) and meridional wind (V), in units of m $s^{-1}$. The dashed lines show the initial profiles used in the WRFLES experiments. The dashed line overlying water vapor mixing ratio is the initial profile for the total water mixing ratio.

**Figure 3**: Time series from runs varying CCN concentrations with "constant" ice formation. a) LWP, in units of $gm^{-2}$. b) IWP, in units of $gm^{-2}$. c) Scatterplot of minimum longwave heating rate vs LWP, in units of K $hour^{-1}$ and g $m^{-2}$, respectively. d) Scatterplot of integrated buoyancy vs LWP, in units of $m^3 s^{-3}$ and g $m^{-2}$, respectively. For INP factors equal to 1.0 and CCN factors 2.0 (blue), 1.0 (black), and 0.5 (red).

**Figure 4**: Impact of increasing CCN by 2X (ConIce2.0 minus ConIce1.0) on (a) relative humidity, (b) sublimation rate, (c) water vapor mixing ratio, and (d) temperature, in units of percent, g $m^{-3}$ $s^{-1}$, g $kg^{-1}$, ℃, respectively.

**Figure 5**: Time series from runs varying CCN concentrations with "constant" ice formation. a) Maximum effective radius, in units of microns. b) Cloud mixed-layer liquid-ice water static energy, in units Kelvin. c) Cloud base ice plus snow number concentration, in units of $L^{-1}$. d) Vertically integrated net deposition rate, in units of g $m^{-2}$ $day^{-1}$. For CCN factors 0.5 (red), 1.0 (black), and 2.0 (blue). For INP factors equal to 1.0 and CCN factors 0.5 (red), 1.0 (black), and 2.0 (blue).





**Figure 6**: Time series from runs varying CCN concentrations with "linear" ice formation. a) LWP, in units of gm$^{-2}$. b) IWP, in units of gm$^{-2}$. c) Liquid water fraction for ConIce (solid) and LinIce (dashed). For CCN and INP factors 0.5 (red), 1.0 (black), and 2.0 (blue).

**Figure 7**: Time series from runs varying CCN concentrations with "linear" ice formation. a) Maximum effective radius, in units of microns. b) Cloud mixed-layer liquid-ice water static energy, in units Kelvin. c) Cloud base ice plus snow number concentration, in units of L$^{-1}$. d) Vertically integrated net deposition rate, in units of g m$^{-2}$ day$^{-1}$. For CCN and INP factors 0.5 (red), 1.0 (black), and 2.0 (blue).

**Figure 8**: (a) CCN and (b) INP in "linear" ice runs at hour 10, in units of cm$^{-3}$ and L$^{-1}$, respectively. For CCN and INP factors 0.5 (red), 1.0 (black), and 2.0 (blue).

**Figure 9**: Impact of prognostic CCN (LinIce2.0-FixCCN2.0) over hour 10. a) Droplet number concentration (black) and snow mixing ratio (red), in units of cm$^{-3}$ and 1.e$^{4}$*g kg$^{-1}$, respectively. b) Longwave radiative heating rate, in units of K hour$^{-1}$. c) Liquid water content, in units of g m$^{-3}$. d) Buoyancy, in units of m$^2$ s$^{-3}$.



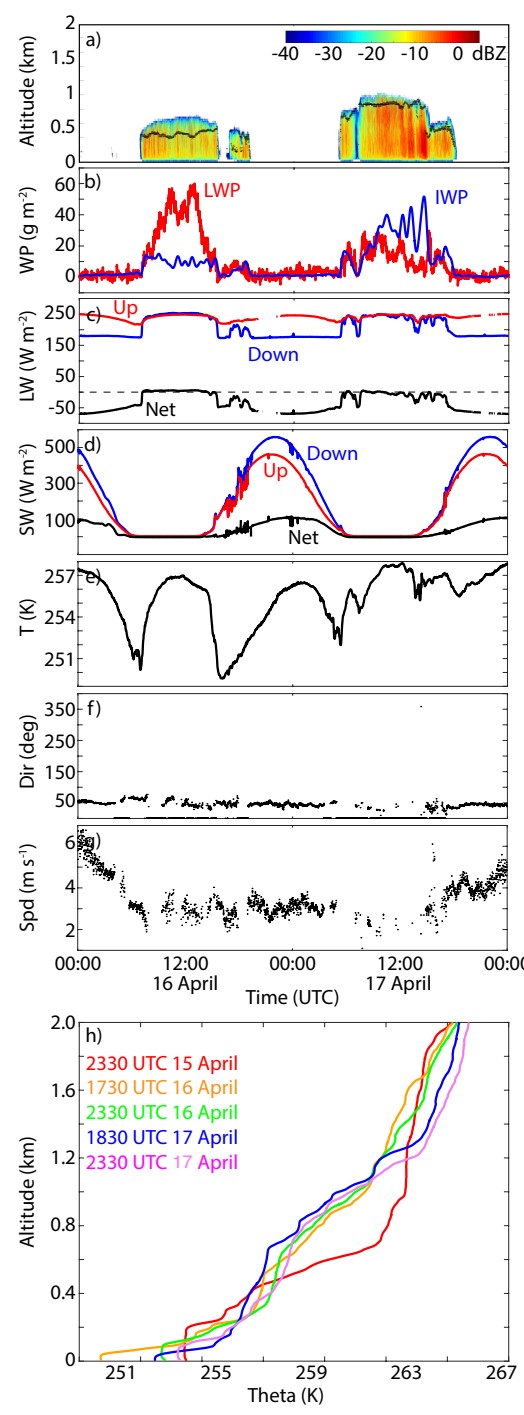





**Figure 1**: Measurements at Oliktok Point, Alaska 15-17 April 2015. (a) Ceilometer (black
dots) and Ka-band cloud radar reflectivity (colored shading). B) LWP and IWP, in units of
gm$^{-2}$. LW (c) and SW (d) surface radiation measurements, in units of Wm$^{-2}$. (e) Surface air
temperatures. (f) Surface wind direction, in units of degrees. (g) Surface wind speed, in units
of ms$^{-1}$. (h) Radiosondes at approximately 2330 UTC on 15 April, 1730 UTC and 2330 UTC
on 16 April, and 1830 UTC and 2330 UTC on 17 April.





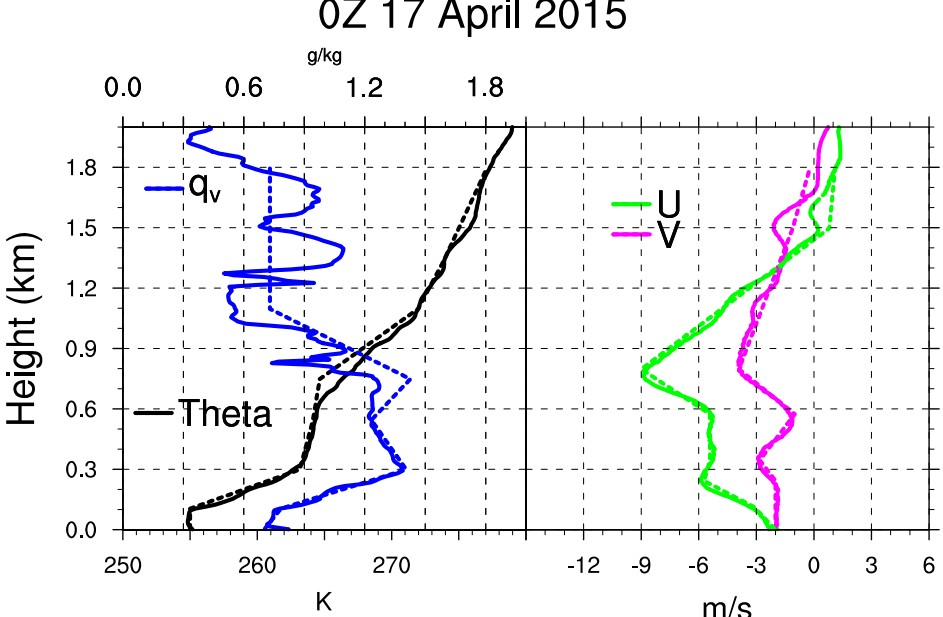

**Figure 2**: Sounding measured at 0 UTC 17 April 2015 at Oliktok Point, Alaska. Left: water
vapor mixing ratio ($q_v$) and potential temperature (theta), in units of g kg$^{-1}$, and Kelvin
respectively. Right: zonal wind (U) and meridional wind (V), in units of m s$^{-1}$. The dashed
lines show the initial profiles used in the WRFLES experiments. The dashed line overlying
water vapor mixing ratio is the initial profile for the total water mixing ratio.





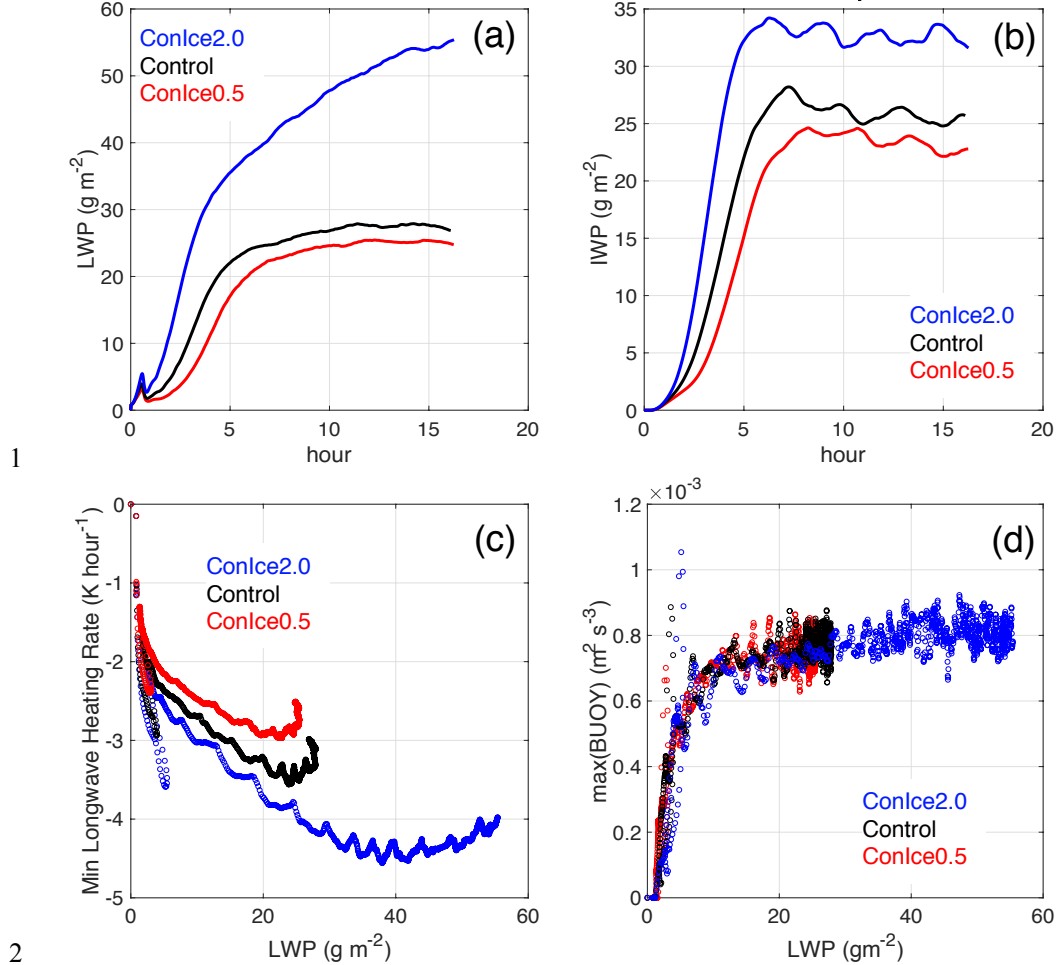

**Figure 3**: Time series from runs varying CCN concentrations with "constant" ice formation.
a) LWP, in units of $gm^{-2}$. b) IWP, in units of $gm^{-2}$. c) Scatterplot of minimum longwave heating
rate vs LWP, in units of K hour$^{-1}$ and $g\,m^{-2}$, respectively. d) Scatterplot of integrated buoyancy
vs LWP, in units of $m^3s^{-3}$ and $g\,m^{-2}$, respectively. For CCN factors 2.0 (blue), 1.0 (black), and
0.5 (red).





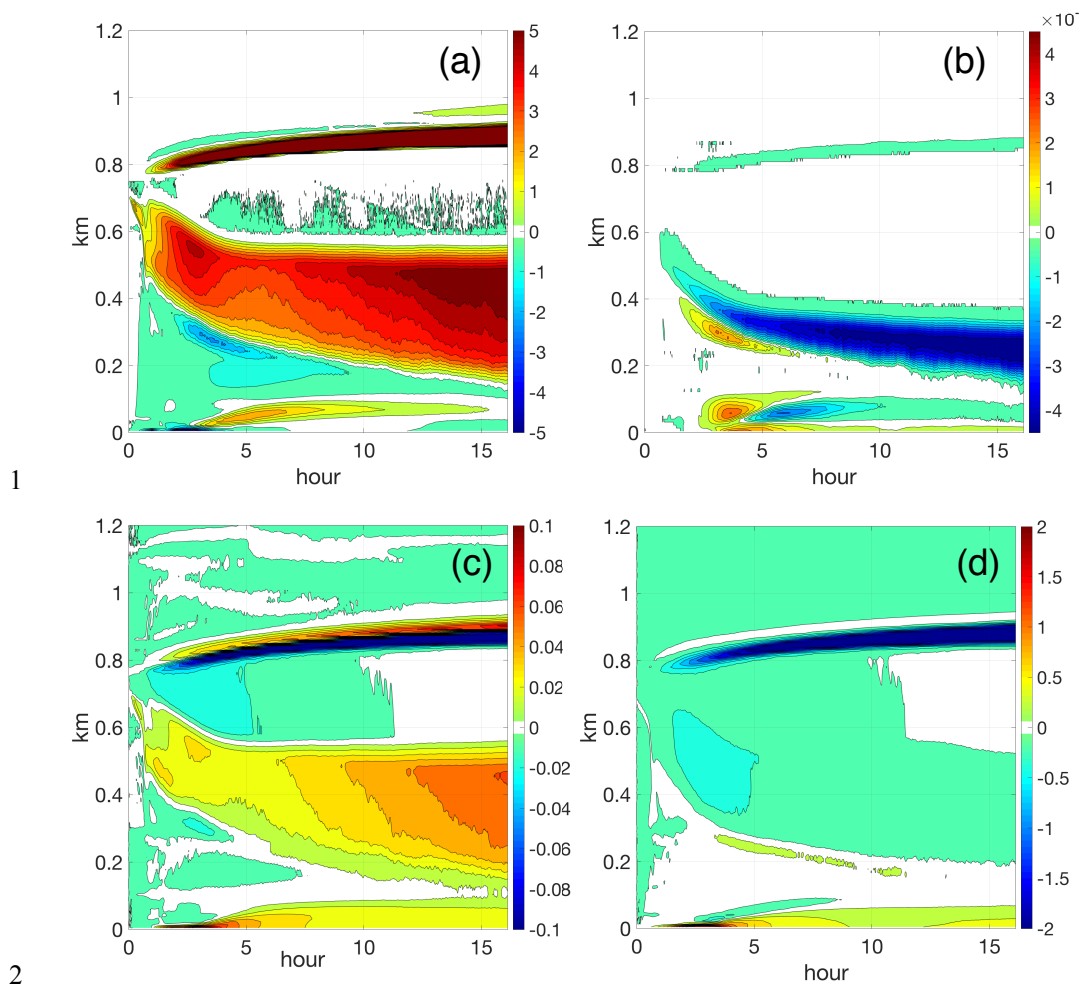

**Figure 4**: Impact of increasing CCN by 2X (ConIce2.0 minus ConIce1.0) on (a) relative
humidity, (b) sublimation rate, (c) water vapor mixing ratio, and (d) temperature, in units of
percent, g m$^{-3}$ s$^{-1}$, g kg$^{-1}$, $^{o}$C, respectively.



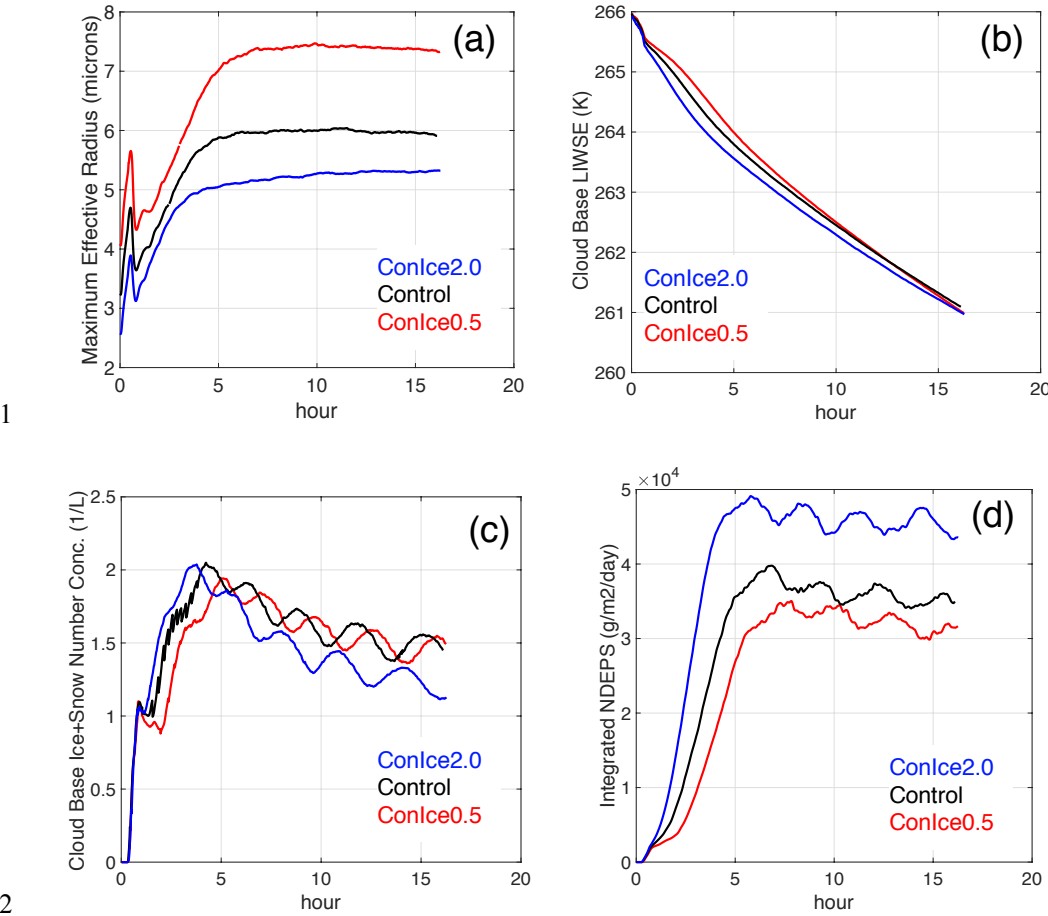

**Figure 5**: Time series from runs varying CCN concentrations with "constant" ice formation.
a) Maximum effective radius, in units of microns. b) Cloud mixed-layer liquid-ice water static
energy, in units Kelvin. c) Cloud base ice plus snow number concentration, in units of $L^{-1}$. d)
Vertically integrated net deposition rate, in units of g $m^{-2}$ $day^{-1}$. For CCN factors 0.5 (red), 1.0
(black), and 2.0 (blue).



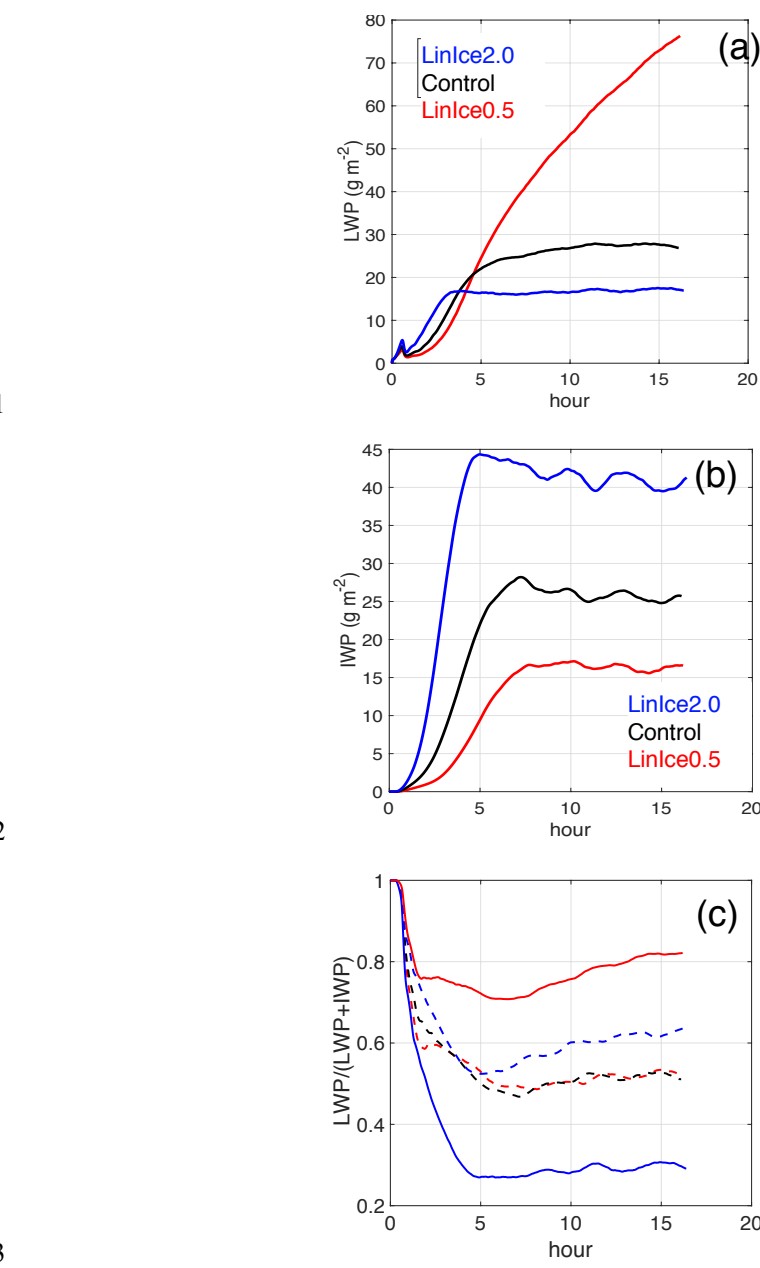

**Figure 6**: Time series from runs varying CCN concentrations with "linear" ice formation. a) LWP, in units of gm$^{-2}$. b) IWP, in units of gm$^{-2}$. c) Liquid water fraction for ConIce (dashed) and LinIce (solid). For CCN and INP factors 0.5 (red), 1.0 (black), and 2.0 (blue).



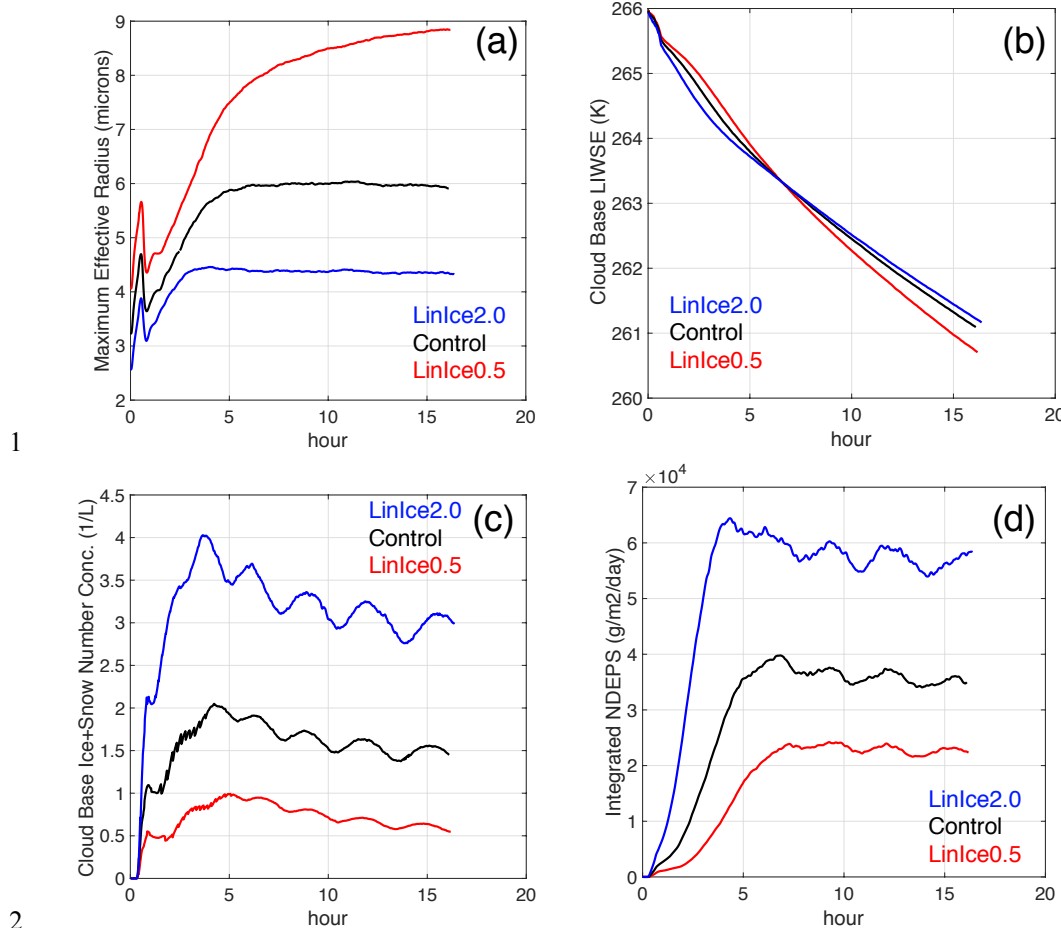

**Figure 7**: Time series from runs varying CCN concentrations with "linear" ice formation. a)
Maximum effective radius, in units of microns. b) Cloud mixed-layer liquid-ice water static
energy, in units Kelvin. c) Cloud base ice plus snow number concentration, in units of $L^{-1}$. d)
Vertically integrated net deposition rate, in units of g $m^{-2}$ $day^{-1}$. For CCN and INP factors 0.5
(red), 1.0 (black), and 2.0 (blue).





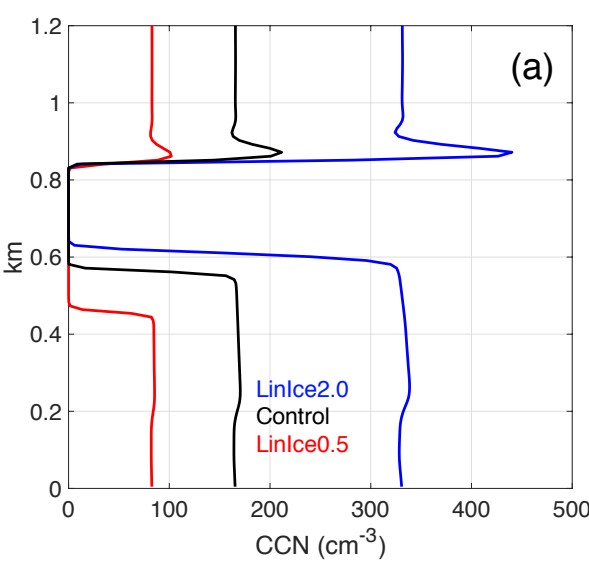

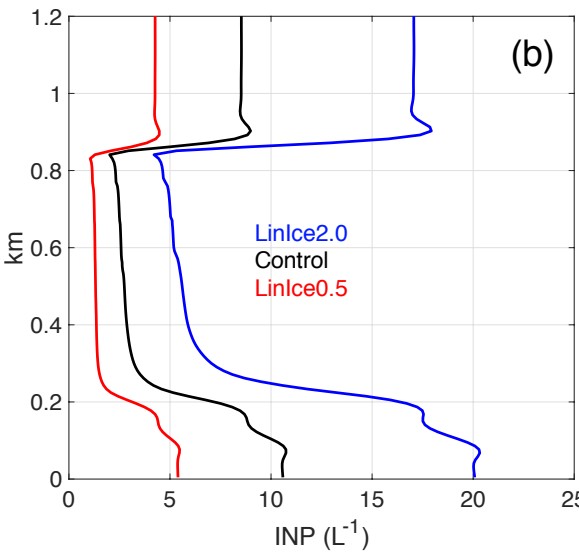

3  **Figure 8**: (a) CCN and (b) INP in "linear" ice runs at hour 10, in units of cm$^{-3}$ and L$^{-1}$,
4  respectively. For CCN and INP factors 0.5 (red), 1.0 (black), and 2.0 (blue).


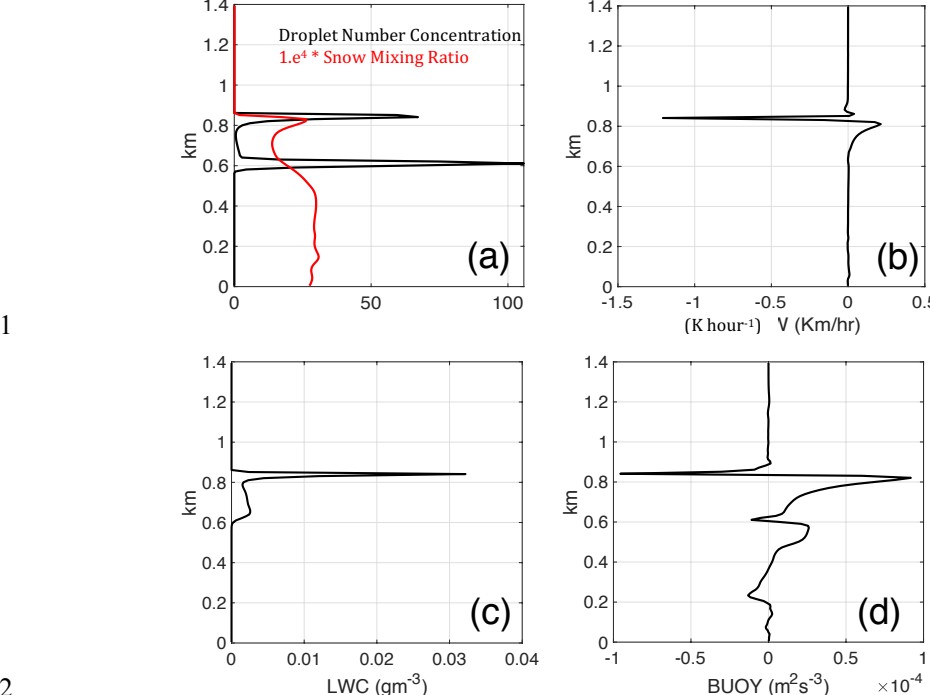

3
**Figure 9**: Impact of prognostic CCN (LinIce2.0-FixCCN2.0) over hour 10. a) Droplet
number concentration (black) and snow mixing ratio (red), in units of $cm^{-3}$ and $1.e^4 \cdot g\ kg^{-1}$,
respectively. b) Longwave radiative heating rate, in units of K $hour^{-1}$. c) Liquid water
content, in units of g $m^{-3}$. d) Buoyancy, in units of $m^2\ s^{-3}$.