# Peer review of "The relative impact of cloud condensation nuclei and ice nucleating"

_Atmospheric Chemistry and Physics, 2018_

## Referee Comment (RC1) · Anonymous Referee #1 · 26 Sep 2018

**Review for "The relative impact of cloud condensation nuclei and ice nucleating particle concentrations on phase-partitioning in Arctic Mixed-Phase Stratocumulus Clouds"**

Solomon et al assess the relative impact of CCN and INP perturbations on the cloud properties of Arctic mixed-phase clouds in a numerical study based on observations obtained at Oliktok Point in Alaska for the night of April 16[th] 2015. The authors identify a range of interesting mechanisms in which the cloud response to the aerosol may be buffered in the mixed-phase regime as opposed to the warm-phase regime. Furthermore, their results show that INP perturbations proved to be more efficient at altering cloud properties than CCN perturbations for the same fractional increase/decrease. The study is well-conceived, well written, of interest to the readership of ACP and deserves publication following minor revisions.

Minor Comments:
- P1L26: "decrease in CCN and INPs results in an increase in the cloud-top longwave cooling rate". This statement sounds like the cloud-top cooling rate is getting stronger for a decrease in CCN, which is not the case. I suggest to either refer to it as longwave heating rate at cloud top as done in Fig. 3, or rephrase for clarification.

- P3L2: "only a few INPs are needed to glaciate a cloud". I would argue this still to be an open question (as the authors discuss in their conclusions). As this is relevant to the paper a brief discussion on this issue with a wider referencing of the excisiting literature may be appropriate here. References that come to mind include:

   Loewe et al, ACP, 2017: "Modelling micro- and macrophysical contributors to the dissipation of an Arctic mixed-phase cloud during the Arctic Summer Cloud Ocean Study (ASCOS)"

   Stevens et al, ACP, 2018: "A model intercomparison of CCN-limited tenuous clouds in the high Arctic"

- P10L11: "LWP consistent with observations": From the observations it seems that a considerably thicker MPC develops during the night with LWP reaching 60 g/m$^2$ and values above 40g/m$^2$ for what seems like ~6h of the night. The LWP in the simulations seems understimated. Please comment.

- The layering of the CCN and INP when a prognostic treatment is used is an interesting finding of this study. Can the authors elaborate why they believe turbulent transport to cause the build-up of CCN above the cloud? Would one not expect turbulent mixing of cloud droplets out of the cloud to be similarly efficient to entrainement of CCN from above into the cloud? It seems unclear to me how turbulence could generate a gradient in number concentration? Please elaborate.

- Figure 5: What does Ni+Ns look like at cloud top? Would one not expect changes in Ni here due to the decrease in temperature (Fig. 4d). Here the nucleation of new crystals occurs which then get processed and mixed through the cloud. So while there are no changes in number concentration at cloud base, the increased cloud top cooling may drive changes in ice crystal number concentration elsewhere in the cloud?

- P6L12: Please add for clarification what the remaining pathway for snow formation is in your model? I would assume that only growth of ice crystals by vapour deposition remains?

- The mechanism of CCN and INP changes impacting the LW cloud top cooling rate (even for thick clouds) and the consequent changes in IWP (even for CCN changes only) is very interesting. It had also been found and hypothesised in a different model for a different case by Possner et al. (2017) where an increase in CCN increased cloud-top cooling, which increased the ice crystal number concentration. Ice water mass increased by increased vapour deposition onto the more numerous crystals (similar as to what the authors see in Fig. 7 for their LinIce experiments). While the feedback here manifests itself differently in simulations with CCN seeding only, it is encouraging to see consistency amongst models and different cases where a feedback through cloud-top cooling impacts the ice phase and stabilises the cloud. This may be worth adding to your discussion.

  Possner et al, GRL, 2017: "Cloud response and feedback processes in stratiform mixed phase clouds perturbed by ship exhaust"

- Figure 9a: Consider adding droplet number concentration profile for LinIce2.0 simulation, which is not shown elsewhere in the manuscript, to show how the prognostic treatment of CCN affects the Nd profile.

---

## Referee Comment (RC2) · Anonymous Referee #2 · 5 Oct 2018

Review for "The relative impact of cloud condensation nuclei and ice nucleating particle concentrations on phase-partitioning in Arctic mixed-phase stratocumulus clouds"

I found this paper to be of good quality with significant findings surrounding the relative impact of CCN and INP loadings on the stratocumulus cloud properties. I recommend publication in ACP with minor revisions. Please find some of my thoughts and comments below.

Minor Comments:

[Figure]

Page 3 line 14: Is this due to less big drops ?

Page 3 line 24: ok I think I understand this. The increase in number is due to them not freezing and being removed from the cloud?

Page 3 paragraph 3: I think this could benefit from some discussion of secondary ice particle production mechanisms here even though they aren't particularly relevant to the temperature range in this study.

Page 10 line 19: Evaporated within 200 m of cloud base. 'below cloud base' should be stated.

Page 11 paragraph 1: I found this interesting. A fine balance indeed!

Page 12 line 13: A reduction in sublimation causes more ice to fall out. Why is sublimation varied in this simulation? Reduced sublimation due to the moistening of the layer below the cloud?

Page 14 line 3. This is interesting that you see larger CCN concentrations above the liquid layer in the inversion. Some observations have found an ultra-clean layer above the cloud top. Could you comment on why you might see something different in this case?

Page 14 line 6. Another interesting finding regarding the location of elevated layer of INP lower in the cloud. I wonder what the implications of this could be.

Page 15 line 23 the model physics vs the Morrison study could be of key importance. The clouds are clearly very sensitive to the ice phase and how the various processes are treated is crucial. E.g. sublimation, re-circulation, shielding of the ice from the liquid etc. One paper that may be relevant is Abel et al. (2017). They did some modelling that showed how sensitive stratocumulus clouds in Cold Air Outbreaks were to ice phase processes and part of that was the partitioning between the liquid and the ice.

Page 16: If the cloud becomes coupled to the surface we would lose the reservoir of

INP. Why?

---

## Author Comment (AC1) · 25 Oct 2018

Thank you to both Reviewer #1 and #2 for the thoughtful and constructive comments. Addressing these comments have helped to clarify a number of points in the paper. We have worked to address the comments as completely as possible in the revised manuscript. Point-by-point responses to the comments are below the blue text in black font.

Reviewer #1:

Solomon et al assess the relative impact of CCN and INP perturbations on the cloud properties of Arctic mixed-phase clouds in a numerical study based on observations obtained at Oliktok Point in Alaska for the night of April 16th 2015. The authors identify a range of interesting mechanisms in which the cloud response to the aerosol may be buffered in the mixed-phase regime as opposed to the warm-phase regime. Furthermore, their results show that INP perturbations proved to be more efficient at altering cloud properties than CCN perturbations for the same fractional increase/decrease. The study is well-conceived, well written, of interest to the readership of ACP and deserves publication following minor revisions.

Minor Comments:
1) P1L26: "decrease in CCN and INPs results in an increase in the cloud-top longwave cooling rate". This statement sounds like the cloud-top cooling rate is getting stronger for a decrease in CCN, which is not the case. I suggest to either refer to it as longwave heating rate at cloud top as done in Fig. 3, or rephrase for clarification.
This sentence is referring to the LinIce0.5 run where a similar fractional decrease in INP and CCN produces and increase in LWP. The sentence has been changed to read "…a run with an equivalent fractional decrease in CCN and INPs results in an increase in the cloud-top longwave cooling rate…".

2) P3L2: "only a few INPs are needed to glaciate a cloud". I would argue this still to be an open question (as the authors discuss in their conclusions). As this is relevant to the paper a brief discussion on this issue with a wider referencing of the excisiting literature may be appropriate here. References that come to mind include:

Loewe et al, ACP, 2017: "Modelling micro- and macrophysical contributors to the dissipation of an Arctic mixed-phase cloud during the Arctic Summer Cloud Ocean Study (ASCOS)"
Stevens et al, ACP, 2018: "A model intercomparison of CCN-limited tenuous clouds in the high Arctic"
This discussion is about INP and the studies about glaciation due to limited CCN are outside the scope of this study but these results are very interesting and important so we have reworded the last sentence in this paragraph to read, "However, it is important to note that an environment with a few INP per liter or limited CCN can glaciate a mixed-phase cloud (DeMott et al. 2010; Mauritsen et al. 2011; Loewe et al. 2017; Stevens et al. 2018)."

3) P10L11: "LWP consistent with observations": From the observations it seems that a considerably thicker MPC develops during the night with LWP reaching 60 g/m2 and values above 40g/m2 for what seems like ~6h of the night. The LWP in the simulations seems understimated. Please comment.

Yes, this is the case for the cloud during 16 April. The simulations are being compared to the cloud observed during 17 April, which has LWP less than 30 g/m2 and a liquid fraction less than or equal to 50%.

4) The layering of the CCN and INP when a prognostic treatment is used is an interesting finding of this study. Can the authors elaborate why they believe turbulent transport to cause the build-up of CCN above the cloud? Would one not expect turbulent mixing of cloud droplets out of the cloud to be similarly efficient to entrainement of CCN from above into the cloud? It seems unclear to me how turbulence could generate a gradient in number concentration? Please elaborate.

Yes, it is interesting that the increase in CCN due to turbulent transport can exceed the loss due to entrainment. Thank you for pointing out that we need to add more to explain this result. This result is because the resolved cloud-driven eddies mix the aerosols and hydrometeors at cloud top and the subgrid mixing mix the aerosols into the inversion. This subgrid mixing is due to the weak inversion and mixes the CCN above the entrainment zone. We have added these details to the text.

5) Figure 5: What does Ni+Ns look like at cloud top? Would one not expect changes in Ni here due to the decrease in temperature (Fig. 4d). Here the nucleation of new crystals occurs which then get processed and mixed through the cloud. So while there are no changes in number concentration at cloud base, the increased cloud top cooling may drive changes in ice crystal number concentration elsewhere in the cloud?

Within the cloud layer ice crystals are well-mixed so the ice plus snow number concentration is constant and Ni+Ns at cloud top looks very similar to Ni+Ns at cloud base. There are three sources for INP that can nucleate in the cloud; INP that have been entrainment into the cloud layer at cloud top, activation of additional INP bins as temperatures decrease, and INP advected in from below the cloud layer that have been recycled. These INP activate at cloud top so Ni is largest at cloud top and Ns is largest at cloud base.

6) P6L12: Please add for clarification what the remaining pathway for snow formation is in your model? I would assume that only growth of ice crystals by vapour deposition remains?

Yes this is correct. We have added the sentence, "Therefore, snow water content is dependent on vapor deposition, auto-conversion from ice to snow, and sublimation."

7) The mechanism of CCN and INP changes impacting the LW cloud top cooling rate (even for thick clouds) and the consequent changes in IWP (even for CCN changes only) is very interesting. It had also been found and hypothesised in a different model for a different case by Possner et al. (2017) where an increase in CCN increased cloud-top cooling, which increased the ice crystal number concentration. Ice water mass increased by increased vapour deposition onto the more numerous crystals (similar as to what the authors see in Fig. 7 for their LinIce experiments). While the feedback here manifests itself differently in simulations with CCN seeding only, it is encouraging to see consistency amongst models and different cases where a feedback through cloud-top cooling impacts the ice phase and stabilises the cloud. This may be worth adding to your discussion. Possner et al, GRL, 2017: "Cloud response and feedback processes in stratiform mixedphase clouds perturbed by ship exhaust"

This is very interesting and we thank the reviewer for this comment. In the Summary and discussion section we have added, "Results consistent with this study were found in Possner et al. (2017), which investigated the impact of ship emissions on mixed-phase stratocumulus observed during M-PACE. Possner et al. found increased CCN increased cloud top cooling, which increased ice mass due to vapor deposition, resulting in a decrease in vapor available for droplet formation. These results are consistent with the ConIce simulations, where an increase in CCN caused an increase in ice water content, primarily due to increased vapor deposition."

8) Figure 9a: Consider adding droplet number concentration profile for LinIce2.0 simulation, which is not shown elsewhere in the manuscript, to show how the prognostic treatment of CCN affects the Nd profile.

Thank you for this comment. We have chosen to add these details to the text instead. In Section 5c we have added, "Cloud droplet concentrations are largest approximately 100 meters above cloud base. At hour 10, the maximum cloud droplet concentration at 650 meters is 330 $cm^{-3}$. Figure 9a shows that allowing for prognostic CCN causes sharper cloud base and cloud top droplet concentrations."

Reviewer #2:

I found this paper to be of good quality with significant findings surrounding the relative impact of CCN and INP loadings on the stratocumulus cloud properties. I recommend publication in ACP with minor revisions. Please find some of my thoughts and comments below.

Minor Comments:

1) Page 3 line 14: Is this due to less big drops?
   Yes, the smaller drops reduce the collision-coalescence and make ice nucleation less efficient.

2) Page 3 line 24: ok I think I understand this. The increase in number is due to them not freezing and being removed from the cloud?
   Yes, this is a correct understanding of the Cziczo et al. (2009), Sullivan et al. (20100, Girard and Sokhandan (2014) studies.

3) Page 3 paragraph 3: I think this could benefit from some discussion of secondary ice particle production mechanisms here even though they aren't particularly relevant to the temperature range in this study.
   We appreciate this comment but have chosen to leave the text unchanged since adding this discussion would take the focus away from the main points of this study.

4) Page 10 line 19: Evaporated within 200 m of cloud base. 'below cloud base' should be stated.
   Text changed as suggested.

5) Page 11 paragraph 1: I found this interesting. A fine balance indeed!

This is very interesting and will be a focus of our future work.

6) Page 12 line 13: A reduction in sublimation causes more ice to fall out. Why is sublimation varied in this simulation? Reduced sublimation due to the moistening of the layer below the cloud?
Yes, this is correct. Increased CCN causes larger ice water content due to increased vapor deposition which moistens the air as ice sublimates below the cloud-driven mixed layer. The causes more ice (INP) to fall to the ground and become unavailable for recycling.

7) Page 14 line 3. This is interesting that you see larger CCN concentrations above the liquid layer in the inversion. Some observations have found an ultra-clean layer above the cloud top. Could you comment on why you might see something different in this case?
The efficacy of these processes is dependent on the strength of the inversion. Stronger stratification in the inversion would limit the mixing from the cloud layer into the inversion. In this Section we have added, "This turbulent transport is due to both resolved cloud-driven eddies, which mix the aerosols and hydrometeors to cloud top, and unresolved subgrid mixing, which mixes the aerosols into the inversion away from entrainment. This subgrid mixing into the inversion is due to the weak inversion at cloud top."

8) Page 14 line 6. Another interesting finding regarding the location of elevated layer of INP lower in the cloud. I wonder what the implications of this could be.
We need to investigate this for a range of environmental conditions since the net impact will depend on whether the cloud system is coupled to the surface, the strength of the surface fluxes, whether the cloud system is rising or lowering, the strength and humidity of the inversion, whether shortwave radiation is significant, etc.

9) Page 15 line 23 the model physics vs the Morrison study could be of key importance.
Yes, this is clearly important. We will definitely look into this in detail in follow-on studies.

10) The clouds are clearly very sensitive to the ice phase and how the various processes are treated is crucial. E.g. sublimation, re-circulation, shielding of the ice from the liquid etc. One paper that may be relevant is Abel et al. (2017). They did some modelling that showed how sensitive stratocumulus clouds in Cold Air Outbreaks were to ice phase processes and part of that was the partitioning between the liquid and the ice.

11) Page 16: If the cloud becomes coupled to the surface we would lose the reservoir of INP. Why?
In an uncoupled system, there is a reservoir of INP due to sublimation. Coupling causes the cloud-driven mixed layer to be well-mixed all the way to the surface. This causes the reservoir to be mixed into the cloud layer where the INP activate and then fall out of the cloud system.